# Finterp: Cost-Time Analysis of Video Action Recognition using the Black Scholes Model

## Abstract

We present a novel method to analyze the earliest instant of time at which a pretrained video action recognition neural network is capable of predicting the action class, with high confidence. We exploit the fact that this problem bears similarities with pricing options in a European stock market, consequentially, our approach, Finterp , is inspired by the Black Scholes model in finance. We formulate analogies between the conceptualization of the variables involved in the Black Scholes formula and video frames to derive the appropriate algorithm. We use Finterp to extensively analyze the prediction capabilities of the neural network over time, on multiple diverse datasets. Finterp reveals that optimal frames are concentrated at low instants of time for datasets with scene bias and mid instants of time for datasets with motion bias. We demonstrate that Finterp does not compromise on the confidence of action prediction in an attempt to minimize the length of video observed. The 'Black Scholes Accuracy' for state-of-the-art 3D CNNs such as I3D and X3D stands at $81 - 86\%$, $64\%$ and $25\%$ for Kinetics, UAV Human and Diving-48 respectively, revealing the need to develop neural networks that can learn unique temporal signatures for various actions. Finally, we extend Finterp to make optimal time instant predictions at the hierarchical level, where similar action classes are grouped together, and show that the optimal time instant predictions are at earlier time instants than the corresponding predictions without hierarchy. We will make all code publicly available.

## 1 Introduction

In tasks such as action recognition Carreira & Zisserman (2017), early action recognition Wang et al. (2019), fine-grained action recognition Shao et al. (2020), etc, it is important for neural networks to learn unique temporal signatures Meng et al. (2019); Lim et al. (2021) for different action classes. The development of newer methods towards this goal requires a deep understanding of the decision making of pretrained action recognition models. One facet of such an analysis involves studying Huang et al. (2018) the motion bias of 3D CNNs by altering the temporal distribution and removing critical frames. Another facet is to analyze Price & Damen (2020) the contribution of individual frames towards the final action prediction of the neural network.

Investigating the action prediction capabilities Fernando et al. (2015); Wang et al. (2016); Soo Kim & Reiter (2017); Dong et al. (2017) of pre-trained action recognition 3D CNNs on partially observed videos can impart crucial insights into the temporal knowledge learnt by action recognition neural networks. This has been relatively unexplored in prior work. Consequently, in the context of video action recognition, one of the key issues is 'cost-time analysis'. We define cost-time analysis as finding the *earliest* instant of time at which the action depicted in the video can be predicted correctly with *high confidence*. In order to find the optimal time instant, we need to estimate the cost of predicting action at a specific instant of time. This can be modeled as a function of the proportion of video observed and the confidence of action prediction Schindler & Van Gool (2008).

**Main Contributions:** In this paper, we present a novel method for cost-time analysis, Finterp , using techniques from economics and finance. The frames of a video form a time-series, we consider the video itself as a 'stock' or 'asset' that we wish to predict at best possible cost. Consequently, we postulate that the

cost of action prediction at any given time instant should be a function of (i) the information contained in frames prior to the time instant under consideration, (ii) future frames, (iii) confidence of prediction at various time instants, (iv) variability in prediction as different frames are considered and (v) cost incurred by observing frames upto the time-instant under consideration.

The prediction made at the end of the video determines the conclusive prediction that can be made by observing the entire video or the final price of the 'asset'. This, along with the ground-truth and action prediction made using the partially observed video, enables us to determine if the action in the video can be recognized using the partial/ full video. This description is synonymous to a customer choosing to reserve an asset or stock at an early instant of time (at the corresponding price) by paying a premium, with an option to buy it at its expiry, at either the reserved price (determined at the earlier time instant) or the final price. The customer may also choose to not buy the stock at all (related to the case when neither the prediction using the partial video nor the final prediction are accurate). This interpretation is reminiscent of the European call option Black & Scholes (2019). The Black-Scholes model Merton (1976), a Nobel prize winning conceptualization, has been widely used in financial markets to price European call options and we use it for cost-time analysis. In particular, the contributions of our work include:

- We present a novel algorithm, Finterp , built on the Black Scholes model, for cost-time analysis i.e. finding the earliest instant of time at which pretrained action recognition neural networks are capable of predicting the action class with high confidence. We carefully design the cost function and the variables for the Black Scholes formula, and show that the assumptions made by the Black Scholes model largely hold for our problem.

- We extensively analyze the prediction capabilities of Finterp over time on diverse datasets spanning front-view, oblique and aerial cameras, low resolution human agents, noisy videos, scene bias and motion bias. Finterp reveals that optimal frames are concentrated at low instants of time for datasets with scene bias and mid instants of time for datasets with motion bias.

- We perform confidence analysis to prove that Finterp does not compromise on the confidence of action prediction in order to minimize the length of video observed.

- We extend Finterp to predict the optimal frame at a hierarchical level Surís et al. (2021); Lan et al. (2014) where 'similar' classes are grouped together. Predictions at the hierarchical level can be made at earlier instants of time than the corresponding predictions without hierarchy.

## 2 Related Work

**Action Recognition.** The availability of large-scale datasets Carreira & Zisserman (2017); Li et al. (2018) has fostered the development of video recognition Singh et al. (2021); Zhu et al. (2020); Simonyan & Zisserman (2014); Kondratyuk et al. (2021) using deep learning. There exist a myriad of methods for action recognition using 3D CNNs Feichtenhofer (2020); Carreira & Zisserman (2017), transformers Yan et al. (2022); Bertasius et al. (2021), for front-camera videos Monfort et al. (2019), ego-recognition Plizzari et al. (2022); Sudhakaran et al. (2019), aerial video recognition Kothandaraman et al. (2022), future action prediction Lan et al. (2014); Surís et al. (2021). More related to our paper is early action recognition Wang et al. (2019); Eun et al. (2020); Pang et al. (2019); Ryoo (2011) where neural networks are specifically trained to be able to recognize actions using partially observed videos. In contrast, the goal of our paper is to analyse pre-trained action recognition models and determine the earliest instant of time they are capable of recognizing the action depicted in the video.

**Understanding video recognition models.** The analysis Samek et al. (2021); Ramakrishnan et al. (2019); Lin et al. (2021); Li et al. (2021a) of spatial information has been facilitated by visualization tools such as attention maps, GradCAM Selvaraju et al. (2016), integrated gradient Qi et al. (2019), etc. Prior work has also investigated issues related to scene bias Byvshev et al. (2022); Hara et al. (2021); Hartley et al. (2022). Temporal analysis Li et al. (2021b) has been explored from the perspective of frame contribution Price & Damen (2020), motion bias Huang et al. (2018) and explicitly modeling Li et al. (2020) of the temporal

dimension. The notion of video frames as a time-series has been used in prior work for forecasting Dai & Li (2009) and prediction Zeng et al. (2021). In this paper, the goal is to analyze the earliest instant of time at which pretrained action recognition networks can predict the action class with high confidence.

## 3 Method

### 3.1 Preliminaries: The Black Scholes Model

In this section, we briefly describe the Black Scholes pricing model Merton (1976) used to determine the price of European call options of assets. In layman terms, in a European call option, an investor can choose to lock the price of an asset at any point in time but can buy the stock (if the investor wants to) only at its expiry. This is irrespective of whether the price of the stock moves in a favorable/ unfavorable manner with time. In order to make decisions on the optimal time instant to buy a stock, investors use the Black Scholes model to predict the price of the stock over time. The 5 variables in the Black Scholes formula are:

1. Underlying stock price $S$ - The underlying stock price is the current price of the asset.

2. Strike price $K$ - Strike price is the cost of the asset at the time of expiry.

3. Time to expiration $t$ - This is the time difference between the current instant and time of expiry.

4. Volatility $\sigma$ - Volatility is the variation in prices of the asset/ the variation in the cost function.

5. Risk free rate $r$ - Risk free rate is the minimum return on an investment when the risks by the investor are zero.

To obtain the Black Scholes cost of purchasing an asset, the spot price $S$ is first multiplied by the standard normal probability distribution function. From this result, to obtain the final cost $C$, the strike price $K$ multiplied by the cumulative standard distribution function is subtracted. Mathematically,

$$SN(d_1) - Ke^{-rt}N(d_2), \tag{1}$$

where

$$d_1 = \frac{log\frac{S}{K} + (r + \frac{\sigma^2}{2})(t)}{\sigma\sqrt{t}}, d_2 = d_1 - \sigma\sqrt{t}. \tag{2}$$

### 3.2 Assumptions

In this section, we describe the assumptions made by the Black Scholes model and their relation to Finterp .

1. No dividends are paid out during the life of an option. This is consistent with the fact that the true prediction of a video can be analysed only after observing the entire video. For example, consider two actions: a person opening a door fully, and a person partially opening the door and closing it. If we observe just the first half of the video, it is hard to predict if the person will proceed to open the door fully or will close it.

2. Market movements are somewhat random. Again, this is true for videos Vondrick et al. (2015). As in the above example, it is equally likely that the person will open the door fully or will close it. Moreover, for cost-time analysis, we assume that we have access to all frames of the video. Hence, we do not try to extrapolate frames (or 'market movements').

3. There are no transaction costs in buying the asset, this assumption is not relevant to our problem statement.

4. The volatility and risk free rate of the underlying asset are known and constant. We calculate the volatility and risk-free rate for each video in accordance with the intuitions behind the variables in financial modeling.

5. The returns of the underlying asset are normally distributed. The amount of knowledge that each frame of a video adds to the cumulative knowledge of the video, when frames are analysed sequentially, first increases, peaks and then decreases, as analyzed in Section 5.

### 3.3 Finterp

In this section, we derive the mathematical formulation of Finterp . Let's assume that the instant of time at which we want to estimate the cost of action prediction is after $x * 100\%$ ($0 \leq x \leq 1$) of the video.

#### 3.3.1 Cost function for strike price and spot price

In this subsection, we describe the cost function for strike price and spot price. The Black Scholes model assumes the cost function to be positive valued throughout, right skewed, some degree of kurtosis (fat tails), similar to a log-normal distribution. The cost in our scenario is a function of the length of video observed $x$ and the confidence of prediction obtained by using video upto time instant $x$. We express the confidence of prediction using the 'gradient norm'. The gradient norm at time instant $x$, $G_x$, of the prediction of the neural network with respect to the ground-truth, provides a holistic estimate of the certainty with which the network makes the prediction. High gradient norms imply that the value of gradients are high which is an indication that the prediction confidence is low. Similarly, low gradient norms implies that the value of gradients are low, an indication that the prediction confidence is high. To compute the gradient norm, the first step is to pass the video frames through the neural network to obtain the final probability distribution $p$. Next, we use the classical action recognition loss function multi-class cross-entropy loss $LCE$ to compute the loss. Backpropagation of the cross-entropy loss through the neural network provides us with the gradients corresponding to all parameters (or weights) of the neural network. The gradient norm $G_x$ is the sum of the $L2$ norms of the gradients corresponding to the weights of the frozen neural network. Note that there is no gradient update anywhere, the CNN is frozen. As more frames of the video are observed by the neural network, naturally, the prediction confidence increases because the neural network contains more information about the video to make an accurate prediction. This is true for any video recognition architecture because the neural network tends to predict the action class with higher accuracy when it has more information about the video. When the prediction confidence increases, the uncertainty decreases. Gradient is a measure that quantifies uncertainty, hence, the gradient decreases as more frames are observed. This implies that the gradient norm is a decreasing function. We hypothesize that the gradient norm decreases at a much faster rate than linear. Hence, the graph of the gradient norm $G_x$ is right skewed.

The total cost $C_x$ is a function of both the gradient norm and the length of video observed. Since we want to simultaneously take into consideration the effect of both factors and want to mimic the score function in the modeling of stock prices, we compute the total cost as $G_x * (1 + x)$. The first term, $G_x \times 1$ is the gradient norm itself, and ensures that the gradient norm/ confidence metric has a considerable contribution in the total cost. The second term $G_x \times x$ is the pro-rata gradient norm and takes into account the proportion of video observed.

$G_x$ is a decreasing function, $1 + x$ is a linearly increasing function. Both $G_x$ and $1 + x$ are always positive, hence $C_x$ is always positive. The product of an increasing and decreasing function first decreases monotonously and then increases monotonously (or the other way around). Since the gradient norm decreases much faster than the rate at which $1 + x$ increases, and has values larger than $1 + x$, the total cost $C_x$ decreases and then increases and is right skewed (similar to vertically inverted log-normal curve). This is consistent with the nature of the score function in the domain of finance. Due to the analogies with assumptions made by the Black Scholes model as well as the stark resemblances between videos and stock market time series, we postulate that the Black Scholes formula can be used to determine the net price of using video frames upto time instant $x$.

#### 3.3.2 Variables for the Black Scholes formula

In this section, we define the variables to be used in the Black Scholes formula (Eq. 1): The spot price $C_{spot}$ is the price of the asset or video at time $x$, computed as $G_x \times (1 + x)$. The final prediction cost which corresponds to the 'price at expiry' (or strike price $C_{strike}$) can be estimated after the entire video has been

observed i.e. $x = 1.0$. Hence, $C_{strike} = G_{x=1.0} \times 2$. The time to expiry $t$ at time instant $x$ is $1 - x$, since $x$ represents the percentage (between 0 and 1) of video observed. Volatility $V$ is a measure of dispersion and is a measure of the variability (of the predictions) of the asset (or video). Entropy is a statistical measure that can provide an estimate of variability. Let $v$ be the softmax predictions of the neural network at time instant $x$, the entropy or volatility $\sigma$ is $\sigma = \Sigma_{\forall} - 1 \times v \times log(v)$. The risk free rate $r$ is the return in the ideal case. In the ideal case, the 'customer' chooses to observe the entire video and also makes the correct prediction. In that case, the probability of the correct class predicted by the neural network using the entire video is the equivalent of the risk free rate. Hence, we define the risk free rate $r$ to be equal to the softmax probability (of the prediction of the neural network) corresponding to the ground-truth label.

We now have all variables required to estimate the price of the asset/ video $C_{BS}$ at time instant $x$ using the Black Scholes formula, as defined in Equations 1 and 2 i.e. $C_{BS} = C_{spot} * N(d_1) - C_{strike}e^{-r*x} * N(d_2); d_1 = \frac{log \frac{C_{spot}}{C_{strike}} + (r + \frac{\sigma 2}{2})(x)}{\sigma\sqrt{x}}, d_2 = d_1 - \sigma\sqrt{x}$. The time instant $x$ at which the price is the lowest (minima) is the optimal frame i.e. it corresponds to the frame (corresponding to the optimal time instant) that the Black Scholes model determines to be suitable for making a high confidence prediction while observing as less video as possible is the optimal frame. The trade-off between the confidence of prediction and length of video observed is defined by the Black Scholes formula. We use the terms optimal time instant, Black Scholes frame and Finterp frame synonymously.

the frame

### 3.4 Finterp at the Hierarchical Level

In this section, we describe an extension of Finterp to make optimal time predictions at a hierarchical level. Action classes typically form a hierarchy Surís et al. (2021). For example, classes 'riding camel' and 'riding elephant' in Kinetics dataset can be broadly classified under one hierarchical class. Similarly, 'drinking beer' and 'tasking beer' (classes in Kinetics), 'Back-15som-05Twis-FREE' and 'Back-15som-15Twis-FREE' (classes in Diving-48) can assigned the same hierarchical label. We wish to analyze the earliest time instant/ frame at which the network can predict the class correctly, up the hierarchy. In hierarchical action recognition, similar classes are grouped together and assigned a common 'hierarchical label'. We analyze (i) if the neural network is able to predict the action class correctly at the hierarchical level even if the accuracy at the fine-grained level is low, and (ii) if the hierarchical prediction can be made at an earlier instant of time as compared to the fine-grained prediction. To do so, we propose a simple modification Finterp described in Section 3.3. Specifically, the variables involved in the Black Scholes formula are modified as follows.

The risk free rate $r$ will remain the same as before as it reflects the ideal case. Volatility $\sigma$ takes into consideration the prediction confidence across all classes, hence, it doesn't change either. Time to expiry $T - i$ is independent of the formulation of action classes. We modify the spot price and strike price as follows. In the event that the prediction is correct, we use the same cost as before. In cases where the fine-grained prediction is incorrect but hierarchical prediction is correct, we compute the gradient norm as the average of the gradient norm with respect to the correct fine-grained action class and the predicted incorrect fine-grained action class.

## 4 A note on the Black Scholes model and the PDE

In this paper, we use the Black Scholes model for cost-time analysis, where the goal is to find the earliest instant of time at which the pretrained action recognition neural network can predict the action class with high confidence. The Black Scholes formula is the solution to a parabolic partial differential equation, derived by the hypothesis that the stock prices follow the geometric brownian motion/ wiener process under certain assumptions, as described in the main paper. In light of this, one option is to treat our problem of cost-time analysis from a mathematical standpoint and derive the appropriate cost function by applying precise assumptions and properties related to the evolution of video frames. Such an approach will allow us to bypass the understanding of various terminologies from finance. However, due to the diversity of videos, it is very difficult to make such a derivation. Hence, instead, we choose to exploit the analogies between

financial markets and our problem of cost-time analysis to derive a suitable solution. While the factors surrounding financial markets and cost-time analysis are not exactly the same, they are similar to a great degree, allowing us to use the Black Scholes model for cost-time analysis. Keeping up with the fact that we rely on the analogies between video cost-time analysis and financial markets to derive the appropriate solution, it is judicious to deduce all the variables in the Black Scholes formula for cost-time analysis as per the intuitions and techniques behind their characterizations in finance.

## 5 Experiments and Results

### 5.1 Datasets, Network and Training Details.

We analyze on a variety of videos with varied camera angles, low-resolution human agents, videos with scene bias and motion bias, and hierarchies. In order to ease analysis and prevent large-scale averaging, we choose small-scale datasets for analysis. Consequently, we test our models on five diverse datasets - three subsets of Kinetics, Diving-48 and UAV Human.

Kinetics is a front camera dataset, Diving-48 is an oblique view dataset and UAV Human is an aerial dataset. Kinetics contains human agents that occupy a large proportion (60% or more) of the video frame. UAV Human and Diving-48 have low-resolution human agents, i.e., the human agents occupy less than 20% of the pixels in the video frame. We curate 3 subsets of classes from the 400 classes in Kinetics-400, each with 20-25 classes. The first set is created such that the hierarchy of classes is based on common action and the objects involved in the action being executed are different. The second set is created such that there is 'actional hierarchy' ('aerobics' and 'zumba'). The third set contains classes that are largely dissimilar. Diving-48 contains 48 fine-grained actions, with large camera motion, moving and intricate background. For interpretability at the hierarchical level, we create groups within set 1 of Kinetics such that same actions being performed on different objects are grouped together. In set-2 of Kinetics, similar action classes such as 'zumba' and 'aerobics' are grouped together. Set-3 of Kinetics has no hierarchy. Hierarchy 1 of Diving-48 groups classes based on the style of diving, a scene-level hierarchy. Hierarchy 2 of Diving-48 groups classes based on the number of somersaults performed, a motion-level hierarchy. We use two popular 3D CNN based action recognition backbones - I3D Carreira & Zisserman (2017) and X3D Feichtenhofer (2020). Please refer to the appendix for more details.

### 5.2 Finterp : Analysis

In this section, we present the results for Finterp . We wish to reiterate that we use the terms optimal time instant, Black Scholes frame and Finterp frame synonymously. The metrics used for analysis are as follows:

1. $x$-Accuracy: For each time instant $x$, the number of videos correctly predicted divided by the total number of videos in the test set is the $x$-accuracy at time instant $x$.

2. Min. Frame Accuracy: This is the ratio of the number of videos for which time instant $x$ is the lowest instant of time at which the prediction has been correctly made to the total number of videos correctly predicted by the network over all time instants.

3. Future correctness percentage: Let the lowest time instant at which the neural network correctly predicts the video is $x = x1$. If at all time instants greater than $x1$, the network correctly predicts the action class depicted in the video, the video is said to be 'future correct'. The ratio of the number of 'future correct' videos to the total number of videos correctly predicted by the networks is the future correctness percentage.

4. Black Scholes Score %: Computed for each time instant $x$, this metric is the ratio of videos that have the best (lowest) Black Scholes score at time instant $x$ as well as are predicted correctly to the total number of videos that have been predicted correctly at time instant $x$. This metric provides the time domain interpretation for optimal frame predictions made by Finterp .

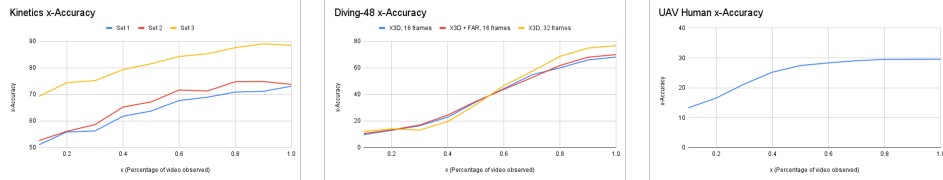

Figure 1: **xAcc trends.** We analyze the prediction capabilities of the model across time. For each time instant $x$, the number of videos correctly predicted divided by the total number of videos in the test set is the $x$-accuracy at time instant $x$. It is an increasing function and the slope is higher for datasets with scene bias than datasets with motion bias.

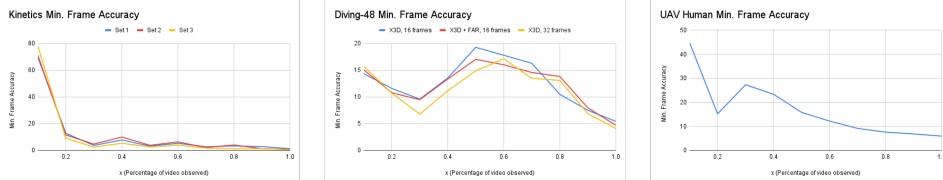

Figure 2: **Min. Frame Accuracy trends.** Min. Frame Accuracy is the ratio of the number of videos for which time instant $x$ is the lowest instant of time at which the prediction has been correctly made to the total number of videos correctly predicted by the network over all time instants. The Min. Frame accuracy peaks at lower values of $x$ for scene-based datasets in comparison to motion-based datasets.

5. Black Scholes Accuracy: Computed for each time instant $x$, this metric is the ratio of videos that have the best (lowest) Black Scholes score at time instant $x$ as well as are predicted correctly to the total number of videos correctly classified by the network at $x = 1.0$. The Black Scholes accuracy denotes the distribution of selected (and accurate) Finterp time instants across time.

6. Black Scholes Optimal Frames %: This metric analyzes the optimal frames predicted by Finterp , across time, including correct as well as incorrect action predictions. At $x$, the Black Scholes optimal frames % is the ratio of the number of videos for which $x$ is the Black Scholes optimal frame to the total number of videos in the test set.

Table 1: Future correctness % validates the accuracy of predictions made after the lowest time instant at which the prediction is correct, datasets with action predictions reliant on motion-bias have a lower future correctness % than those reliant on scene-bias.

| Finterp | |
|---|---|
| Dataset | Future Correctness % |
| Kinetics Set 1 | 76.83 |
| Kinetics Set 2 | 82.13 |
| Kinetics Set 3 | 87.59 |
| UAV Human FAR | 68.1 |
| Diving-48 X3D (16 frames) | 62.88 |
| Diving-48 X3D + FAR (16 frames) | 64.95 |
| Diving-48 X3D (32 frames) | 67.30 |
| Hierarchical Finterp | |
| Kinetics Set 1 | 67.1 |
| Kinetics Set 2 | 73.47 |
| Diving-48 X3D (32 frames) - Hierarchy 1 | 47.71 |
| Diving-48 X3D (32 frames) - Hierarchy 2 | 48.02 |

**x-Accuracy.** (Figure 1). For all datasets, the $x$-Accuracy is an increasing function from time $x = 0.1$ to $x = 1.0$. For Kinetics and UAV Human, the $x$-accuracy at low time instants such as $x = 0.1, 0.2, 0.3$ is close to or much higher than 50% of the $x$-accuracy at $x = 1.0$. This is because in many videos in scene-based

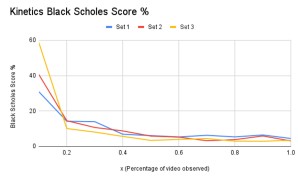 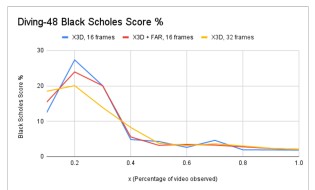 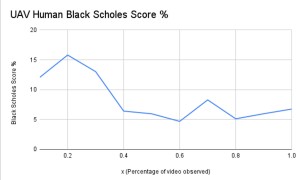

Figure 3: **Black Scholes Score % trends.** Computed for each time instant $x$, this metric is the ratio of videos that have the best (lowest) Black Scholes score at time instant $x$ as well as are predicted correctly to the total number of videos that have been predicted correctly at time instant $x$. This metric provides the time domain interpretation for optimal frame predictions made by Finterp .

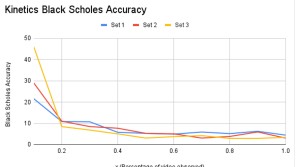 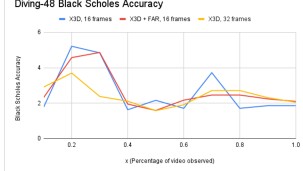 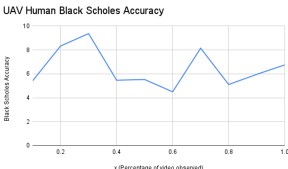

Figure 4: **Black Scholes Accuracy trends.** Computed for each time instant $x$, this metric is the ratio of videos that have the best (lowest) Black Scholes score at time instant $x$ as well as are predicted correctly to the total number of videos correctly classified by the network at $x = 1.0$. The Black Scholes accuracy denotes the distribution of selected (and accurate) Finterp time instants across time.

datasets such as Kinetics and UAV Human, the action can be identified from the scene (or spatial information) and a small amount of temporal information. In contrast, in motion-based datasets such as Diving-48, it is important for the network to observe sufficient amount of temporal information and decipher the action based on the motion executed by the human agent. In some cases, we observe a slight fluctuation of accuracies where accuracies at higher time instants are slightly lower than the accuracies at lower time instants. This is because of ambiguities in the motion in certain future frames corresponding to the time interval between the correct prediction and the incorrect prediction, resulting in the networks' erroneous prediction.

**Min. Frame Accuracy.** (Figure 2). The min. frame accuracy for scene-based dataset Kinetics is high at low values of $x = 0.1$. This is because, when the reliance on spatial content is higher than the reliance on temporal content, prediction can be made early. In contrast, for Diving-48, the min. frame accuracy peaks at mid values of $x = 0.3 - 0.8$. This is because motion-based datasets such as Diving-48 require a lot of temporal information for interpretation. For highly complex datasets such as UAV Human, a major portion of the min. frame accuracy is concentrated around low to mid values $x$. Due to camera angle and limited visibility of scene elements (spatial information), observing more frames (temporal information) helps the neural network identify the action better. The min. frame accuracy is typically low at high values of $x = 0.9, 1.0$, thus, for very few videos, the neural network needs to observe the entire video to make the action prediction.

**Future correctness %.** (Table 1). Future correctness does not talk about the specific instant of time at which the prediction is made, rather, it validates whether all predictions made after the lowest time of correct prediction are correct. If a video is 'future correct', it implies that the neural network is able to use just the first part of the video to accurately model the action class and is not ambiguous when future frames are observed. The future correctness percentage of Kinetics, is very high ($70\% - 90\%$). For Diving-48, a dataset with heavy motion bias, the future correctness percentage is low due to potential inconsistencies in future frames. The future correctness percentage of UAV Human is around $68\%$, the dataset is very complex due to variations in lighting, noise and low resolution human actors.

**Black Scholes Score %.** (Figure 3). The Black Scholes score % signifies the proportion of videos that have the best Black Scholes score at time $x$, given that the classification is correct. For scene-based dataset Kinetics, the network is able to predict correctly with high confidence at low instants of time $x = 0.1, 0.2$.

Figure 5: **Black Scholes optimal frames % trends.** At $x$, the Black Scholes optimal frames % is the ratio of the number of videos for which $x$ is the Black Scholes optimal frame to the total number of videos in the test set. This metric does not take into consideration the correctness of action prediction at the optimal time instant. Hence, it throws light on the prediction capabilities of the model, taking into consideration only the prediction confidence and length of video observed.

Hence, the optimal Black Scholes cost, encompassing prediction confidence as well as proportion of video observed, is high at low instants of time. It decreases as $x$ increases. For motion-based Diving-48 dataset where temporal relevance is high and the challenging UAV Human dataset where it is easier to predict when more data is available, the Black Scholes score % peaks at $x = 0.2, 0.3$ and then gradually decreases. In all cases, the Black Scholes score % is low at high instants of $x$, again revealing that a small portion of the dataset requires access to a large portion of the video to predict accurately with high confidence. The Black Scholes Score % at $x = x1$ is lower than the sum of Min. Frame Accuracy from $x = 0.1$ to $x = x1$ demonstrating that the lowest instant of time at which the prediction is correct is not always the optimal time instant, a high confidence of action prediction is desirable. In general, the overall confidence of prediction increases from $x = 0.1$ to $x = 1.0$. While $x = 0.9, 1.0$ could correspond to the highest confidence, it is important to note that our algorithm trades off cost incurred due to observing video frames and prediction certainity.

**Black Scholes Accuracy.** (Figure 4). For Kinetics, the overall trend is that the Black Scholes accuracy is a faster than exponential long-tailed decreasing function. Set-3 has no hierarchy in its classes and has a higher Black Scholes accuracy at low values of $x$, implying that there are few to no confusing action classes, hence, the action can be largely recognized with just a few frames. In other words, recognizing actions by relying on scene-bias works here. The first set has hierarchy based on the object on which action is being performed, and the second set has similar looking actions. Consequently, the second set benefits more from scene bias than the first set. This implies that, at lower values of $x$, we expect the Black Scholes accuracy of set 2 to be higher than the Black Scholes accuracy of set 1. The graph indicates that this is indeed true.

For UAV Human, the network doesn't benefit much from scene bias due to low-resolution human agents. Hence, motion is an important indicator of action being performed. However, since the dataset does not contain too many similar actions, a robust neural network should learn distinct temporal signatures for various actions. Hence, we expect the network to be able to recognize actions using the temporal information from a part of the video, i.e. the overall trend should be such that the Black Scholes accuracy first increases and then decreases. The graph indicates that this is largely true. We notice two peaks, which can be explained by the fact that UAV Human contains videos taken at varying altitudes and angles. Videos captured at relatively lower altitudes and oblique angles can be recognized using a smaller portion of the video. However, videos captured at high altitudes and aerial angles require a larger portion of the video to be passed through the neural network to obtain high confidence predictions.

While Diving-48 contains easily recognizable scene information, it is a motion-centric dataset with a large number of similar actions. For instance, predicting the number of somersaults performed by the human actor requires the neural network to learn unique temporal signatures for all actions. Due to high temporal bias, with varying complexities in videos depending on diving style, the trends are similar to that of UAV Human, the curves have multiple peaks.

Across $x = 0.1...1.0$, the cumulative Black Scholes accuracy is 81.01%, 82.54%, 86.72% for Kinetics Set 1, Set 2 and Set 3 respectively. Thus, at the optimal frame predicted by Finterp , the prediction of action class is accurate for $81 - 86\%$ of Kinetics videos. For the challenging UAV dataset, the Black Scholes accuracy is 64.63%. For Diving-48, the Black Scholes accuracies are low, at 26.52%, 24.42% and 26.67% for the three

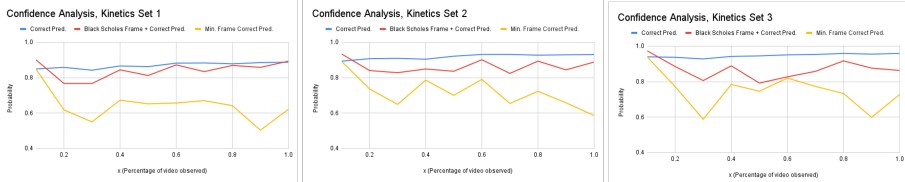

Figure 6: **Confidence analysis on Kinetics.** We present the confidence (probability) analysis of predictions made by the Black Scholes model on the Kinetics dataset and compare it with the overall average probability of predictions and Min. Frame probability of predictions. We demonstrate that the Black Scholes algorithm chooses optimal frames where the softmax probability of (accurate) action class is high, while minimizing the length of video observed.

cases. Finterp trades off length of video observed and confidence of action prediction. During optimization, it uses the confidence of prediction at various time instants (including at the time instant under consideration for optimal frame and $x = 1.0$). Hence, it ensures that the confidence of prediction at the chosen optimal time instant is sufficiently close to the confidence of prediction at $x = 1.0$, while minimizing the length of video used for action prediction. Inaccuracies in action predictions at the Finterp optimal frame motivates the development of neural network architectures that learn unique temporal signatures for different action classes.

**Black Scholes Optimal Frames %.** (Figure 5) The trends are similar to that of Black Scholes accuracy, with similar intuitions and analysis. To estimate the optimal frame, Finterp takes into consideration the proportion of video observed, the confidence at each time instant as well as the final confidence of prediction (at $x = 1.0$). Hence, it makes sure to trade-off the length of video observed, while choosing the optimal instant at a time where the prediction confidence is high and close to the maximal confidence of prediction that can be obtained (at $x = 1.0$, since, prediction confidence, related to $x$-Accuracy is a monotonously increasing function). At instances where the confidence of predictions is high and the action predictions at the Black Scholes optimal time instant is inaccurate, it can be implied that the neural network has not learnt temporal signatures for various action classes in the dataset. When the confidence of predictions is not high, inaccurate action predictions at the Black Scholes optimal time instant reveal either that (i) Finterp s' way of fairly trading off length of video observed and confidence of prediction is incorrect for the specific video, and that the neural network will benefit by observing a larger segment of the video (especially for datasets with highly motion-centric actions, with motion-sensitive fine-grained (similar) actions) or (ii) that the neural network has not learnt feature representations in a robust manner and needs improvements in order to make high confidence predictions.

Finterp determines the optimal time instant mathematically, as per proven approaches in finance and based on the similarities between stock markets and cost-time analysis. Hence, the insights gained from optimal time instant predictions of the Black Scholes model can be used to develop neural network architectures that learn unique temporal signatures for accurate high confidence action predictions at the Black Scholes optimal time instant. Such an approach can be useful for various problem settings including fine-grained action recognition, early action recognition, online action recognition, etc.

### 5.2.1 Confidence Analysis

In this section, we present the confidence analysis for Finterp (Figure 6 and Figure 7). We use the following metrics for analysis:

1. Probability of correct prediction (shown as Correct Pred. on graph) - At $x$, the average probability of prediction of videos classified correctly.

2. Probability of correct Black Scholes prediction (shown as Black Scholes Frame + Correct Pred. on graph) - At $x$, the average probability of prediction of videos classified correctly, where $x$ is also the Finterp frame for the video.

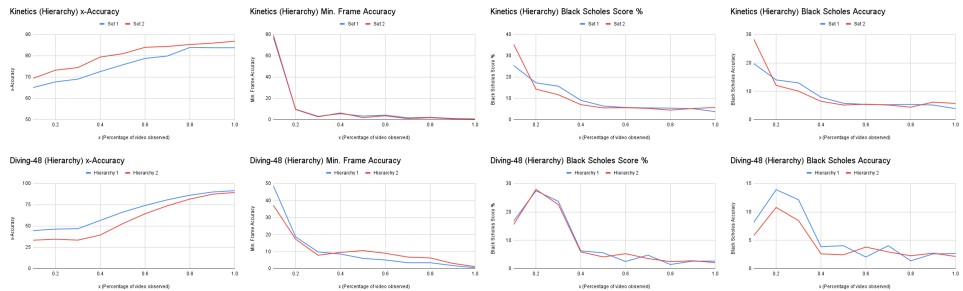

Figure 7: **Confidence analysis on Diving-48 and UAV Human.** We present the confidence (probability) analysis of predictions made by the Black Scholes model on the Diving-48 and UAV Human datasets and compare it with the overall average probability of predictions and Min. Frame probability of predictions. We demonstrate that the Black Scholes algorithm chooses optimal frames where the softmax probability of (accurate) action class is high, while minimizing the length of video observed.

Figure 8: **Finterp at Hierarchical level.** Predictions at the hierarchical level correspond to higher accuracies (over all metrics) at lower values of $x$, across scene-based and motion-based hierarchies.

3. Probability of correct minimum frame prediction (shown as Min. Frame Correct Pred. on graph) - At $x$, the average probability of prediction of videos classified correctly, where $x$ is also the lowest time instant at which the action depicted in the video is predicted correctly.

As per the thesis of the Black Scholes model, we expect the probability of correct Black Scholes prediction to be much higher or equal to the probability of correct minimum frame prediction. This is true across all values of $x$ for all sets of the Kinetics dataset. This is mostly true for complex datasets such as UAV Human and Diving-48 as well, with certain exceptions at low values of $x$. The exceptions are due to extensive motion information required to predict actions in many videos in the dataset due to their inherent complexities, while the Black Scholes model is conditioned to trade-off prediction confidence and length of video observed in a fair manner. The confidence of correct Black Scholes prediction is very similar to the probability of correct prediction, indicating that the Black Scholes algorithm doesn't compromise on the prediction probability in an attempt to minimize the length of video observed.

### 5.3 Finterp at Hierarchical Level

. In this section, we present the results for Finterp at the hierarchical level (Figure 8). We use the X3D backbone with a sampling rate of 32 for all Diving-48 experiments. The $x$-Accuracy for predictions at hierarchical level is higher than the corresponding $x$-Accuracy for temporal predictions w/o hierarchy, across datasets and values of $x$. When similar classes are grouped, incorrect predictions of the neural network within a group of classes can still lead to correct hierarchical predictions. Similarly, the Min. Frame Accuracy at low values of $x = 0.1, 0.2$ for hierarchical predictions is higher. As shown in Table 1, the future correctness percentage for hierarchical predictions is lower. This is because the neural network tends to predict the hierarchical class (correctly) at a very early stage before observing sufficient frames, based on scene bias. As more frames (and motion) are observed, there could be temporary inconsistencies in prediction.

The curve of Black Scholes Score % is shifted towards the left for hierarchical predictions compared to the corresponding curve without hierarchy, thus the optimal frame for hierarchical action prediction is at a lower time instant than the optimal frame w/o hierarchy. In other words, the neural network is able to predict the

hierarchical class correctly at an earlier stage than the fine-grained class, with high confidence. The trends in Black Scholes Accuracy for hierarchical predictions are similar to that of predictions made without taking hierarchy into consideration. However, we observe that the absolute values of the Black Scholes Accuracy for hierarchical predictions is higher, even while the x-Accuracy at $x = 1.0$ are not as far apart, indicating that the accuracy at the optimal time instant is higher.

The cumulative Black Scholes Accuracy over $x = 0.1...1.0$ is 55% and 44% for Hierarchy 1 and Hierarchy 2 of Diving-48. The corresponding score for predictions without hierarchy is 26%. This suggests that at the optimal time instant, even for heavy motion-based fine-grained datasets, the action prediction is more accurate when hierarchy is considered. This is because when hierarchy is taken into consideration, fine-grained classes are grouped together and it is less important for the neural network to learn unique temporal signatures for all action classes. Similarly, the cumulative Black Scholes Accuracy over $x = 0.1...1.0$ is 85.47% and 89.03% for Set 1 and Set 2 of Kinetics, as opposed to 81.01% and 82.54% for prediction w/o hierarchy, an improvement of $4\% - 7\%$.

### 5.4 Summary of Analysis

The findings of this paper are as follows:

1. The accuracy of the model increases over time $x$. Scene-based datasets have a high accuracy at low values of $x$, the curve is a decreasing function. For motion-centric datasets, the x-Accuracy curve first increases, peaks and then decreases. When the motion bias is very heavy, there could be multiple peaks. The min. frame accuracy trends are similar.

2. Finterp optimal frames are concentrated towards low values of $x$ for datasets with scene bias, and around mid values of $x$ for datasets with motion bias.

3. The overall Black Scholes accuracy is high for datasets with scene-bias as well datasets with moderate motion bias. Datasets that are fine grained and have high motion bias, the action prediction at the Black Scholes frame is inaccurate for a lot of videos. This motivates the development of neural network that learn unique temporal signatures for different actions.

4. The confidence of predictions at the Black Scholes time instant is significantly higher or equal to the confidence at the min. frame time instant. The confidence of predictions at the Black Scholes time instant is comparable to the average confidence prediction. This indicates that Finterp , while striving to minimize the length of video observed, ensures that the confidence of prediction at the optimal time instant is comparable to the confidence of prediction made using the entire video.

5. Black Scholes predictions at the hierarchical level correspond to higher accuracies (over all metrics) at lower values of $x$, across scene-based and motion-based hierarchies.

## 6 Conclusions

We present the Black Scholes algorithm for an important facet of understanding pretrained action recognition models. A limitation of our method is that it is expensive since it needs to be run on various lengths of the video along with expensive neural network and gradient computations at each step. In this paper, we analyzed only on trimmed videos containing a single action, analysis on untrimmed videos containing multiple action is a direction for future work. Other directions for future work include analysis on more video datasets and models, using the insights gained from our method to develop better algorithms for action recognition and early action recognition, frame selection for sampling, highlight detection, etc.

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

# A  Appendix

## A.1  More details on the assumptions in the Black Model

Relation of Black Scholes assumptions to videos

- Assumption 1: No dividends are paid out during the life of an option. Consider the above example of a person partially opening the door in the frames observed upto time instant i. The entire video needs to be watched to predict the action executed - the person could proceed to fully open the door or close it or leave the door partially opened. While a prediction can indeed be made (upto certain confidence level) at time instant i, the true action can be determined only after watching the video fully. Similarly, in markets, irrespective of when the bid is made, the purchase is done only at the end of the lifetime of the stock. Moreover, 'true prediction' requires observing the entire video - The exact action depicted in the video can be predicted with highest confidence only by observing the

entire video. Prediction at all other stages has lesser probability. For instance, consider a person partially opening a door and then immediately closing it. It is hard to predict the precise action unless the entire video is observed. When there is scene bias, it is possible to predict the action with just a few frames. This is exactly what our paper analyses - our paper analyses how much temporal information is required (and the confidence level as different frames are observed) to make a high confidence prediction.

- Assumption 2: Market movements are somewhat random. Video frames have strong frame-to-frame dependency. Conditioned on the ith frame, the movement of pixel values in the (i+1)th frame (from the ith frame) is random. For instance, consider the case where a person is opening the door in the first i frames. In the subsequent frames, the person could continue to fully open the door, or shut the partially opened door and leave the door partially opened. The action that is executed in the (i+1)th frame is random, conditioned on the ith frame (due to frame-to-frame dependency). Similarly, in a call option in the stock market, the market movement at the (i+1)th time instant is random conditioned on the stock prices at the ith time instant.

- Assumption 3: There are no transaction costs in buying the asset, this assumption is not relevant to our problem statement.

- The volatility and risk free rate of the underlying asset are known and constant. Volatility V is a measure of dispersion and is a measure of the variability (of the predictions) of the asset. In the case of videos, entropy is a statistical measure that can provide an estimate of variability. The risk free rate is the return in the ideal case. In the ideal case, the 'customer' chooses to observe the entire video and also makes the correct prediction. Hence, we define the risk free rate r to be equal to the softmax probability (of the prediction of the neural network) corresponding to the ground-truth label.

- The returns of the underlying asset are normally distributed. We analyze this experimentally in Section 5 and also note that the findings are intuitive with what one would expect with observing video frames.

Note that the method is only applicable in a transductive setting. Through this paper, we are trying to analyze the temporality in videos, which we hope will pave the way for the development of more effective solutions for various video understanding problems in the future.

### A.2 Datasets, Network and Training Details

**Datasets:** We benchmark our models on three datasets - Kinetics, Diving-48 and UAV Human. Kinetics is a front camera dataset, Diving-48 is an oblique view dataset and UAV Human is an aerial dataset. The Kinetics dataset contains human agents that occupy a large proportion (60% or more) of the video frame w.r.t. the background. UAV Human and Diving-48 have low-resolution human agents, i.e., the human agents occupy less than 20% of the pixels w.r.t. the background. The videos in Kinetics were extracted from YouTube and trimmed (temporally) and cropped/ resized (spatially) such that the length of the smaller dimension is 256. For Kinetics, we use a frame rate of 64. We curate 3 subsets of classes from the 400 classes in Kinetics-400, each with 20-25 classes and use the train and test videos corresponding to those sets for experimental analysis. The first set is created such that the hierarchy of classes is based on common action and the object involved in the action being executed. The first set is created such that there is 'actional hierarchy', for e.g. classes 'aerobics', 'zumba' can be classified under the same umbrella class. The third set contains classes that are largely dissimilar with little hierarchical structure.

Diving-48 was compiled by segmenting online videos of diving competitions. It contains 48 fine grained actions, with large camera motion, moving background (oscillating springboard) and intricate background, most of the videos are recorded from oblique angles. We use a frame rate of 16 or 32 frames for analysis, the spatial dimensions are $480 \times 640$.

UAV Human contains low-resolution videos taken under adverse lighting and weather conditions. It has 155 actions, many of which are similar and hard to distinguish. Moreover, the videos contain dynamic background,

| Set-1 |
| --- |
| 47-Catching fish, 48-catching/ throwing baseball, 49catching/throwing frisbee |
| 63-cleaning shoes, 65-cleaning windows |
| 66-climbing rope, 68-climbing tree |
| 81-cutting nails, 83-cutting watermelon |
| 100-drinking, 352-tasting beer, 101-drinking beer |
| 353-tasting food, 110-eating cake, 111-eating carrots,112-eating chips,115-eating icecream |
| 231-playing flute,230-playing drums,250-playing violin |
| 268-riding camel,274-riding scooter,269-riding elephant |

| Set-2 |
| --- |
| 192-marching,281-running on treadmill,168-jogging |
| 95-doing aerobics,399-zumba,398-yoga,84-dancing,85-dancing charleston,86-dancing gangnam style |
| 98-drawing,396-writing |
| 319-sneezing,320-sniffing |
| 304-singing,180-laughing,397-yawning |
| 330-squat,183-lunges |
| 339-swimming backstroke,340-swimming breaststroke,341-swimming butterfly stroke |
| 128-fixing hair,138-getting a haircut |
| 352-tasting beer,101-drinking beer |

| Set-3 |
| --- |
| 5-archery |
| 14-barbequing |
| 42-canoeing or kayaking |
| 91-dining |
| 128-fixing hair |
| 151-high jump |
| 158-hugging |
| 211-peeling potatoes |
| 260-push up |
| 278-rock climbing |
| 286-scuba diving |
| 377-waiting in line |
| 394-wrapping present |
| 398-yoga |

Table 2: **Kinetics-400 Subsets: Classes and Hierarchy.** We create 3 subsets of classes within Kinetics. We list the classes, along with the corresponding Kinetics-400 class labels, in the table above. Classes that share the same hierarchical label are given in the same row.

camera motion and noise. We use video frames with spatial resolution $540 \times 960$ in all our experiments, the frame rate is 8.

For hierarchical early action recognition, set 1 of Kinetics contains same actions being performed on different objects, all clubbed into one hierarchical class. Set-2 of Kinetics has 'actional' hierarchy. Similar classes such as 'zumba' and 'aerobics' are grouped. Set-3 of Kinetics has no hierarchy, hence we we do not perform any hierarchical early action recognition analysis. Hierarchy 1 of Diving-48 groups classes based on the style of diving, a scene-level hierarchy. Hierarchy 2 of Diving-48 groups classes based on the number of somersaults performed, a motion-level hierarchy.

**Classes and Hierarchy.** We list the set of classes using in our Kinetics experiments in Table 2. For diving-48, we use the diving style and number of somersaults information provided by the dataset to create hierarchical labels pertaining to Hierarchy 1 and Hierarchy 2 respectively.

**Training details:** All our models were trained using NVIDIA GeForce 1080 Ti GPUs, and NVIDIA RTX A5000 GPUs. Initial learning rates were set at 0.01, and Randomly Initialized Uniform Sampling was used for frame sampling while training. We use the Stochastic Gradient Descent (SGD) for optimization, with 0.0005 weight decay, and 0.9 momentum. We use cosine/poly annealing for learning rate decay and multi-class cross entropy loss to constrain the final softmax predictions of the neural network. We report top-1 accuracies in all cases.

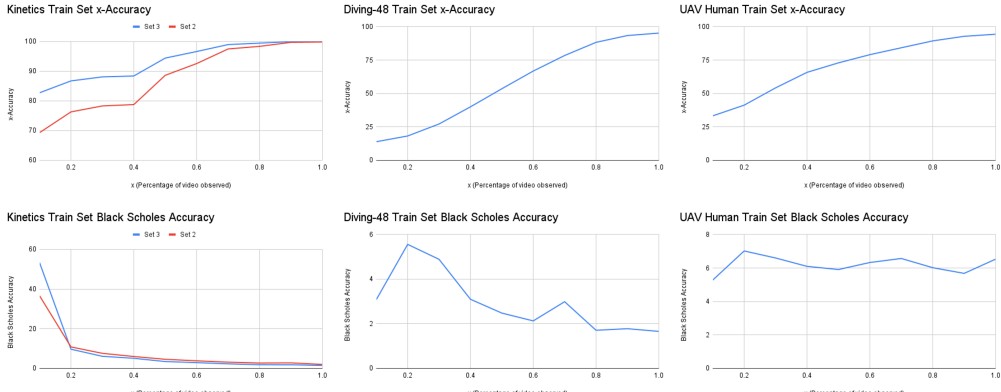

Figure 9: **Analysis on the train set.** We analyze the prediction capabilities, trends and optimal time instant predictions of the Black Scholes temporal interpretability algorithm, across time, on the train set of various datasets.

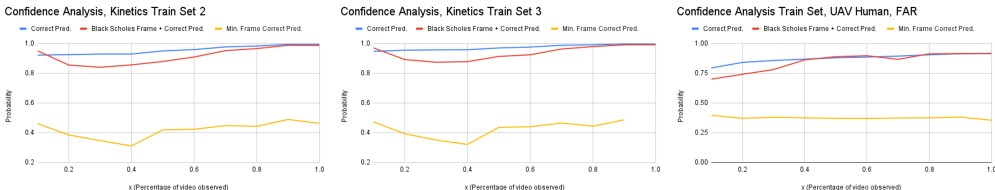

Figure 10: **Confidence analysis on the train set.** Confidence analysis for the train set of various datasets.

### A.3 Analysis on the Train Set

In the preceding sections, we have extensively analysed Finterp on the test set of various datasets. Neural networks, naturally, tend to have a far higher action recognition accuracy on training data. In this section, we analyze Finterp on the train set of various datasets. For Kinetics, we analyze on Set 2 and Set 3 - Set 2 is motion based and set 3 is scene based. For Diving-48, we use the X3D model, trained with a frame rate of 16. For UAV Human, we show results on I3D + FAR.

Please refer to Figure 9 for $x$-Accuracy and Black Scholes accuracy trends. The trends are very similar to that of the test set, indicating that the network as well as Finterp generalize well across the train and test sets. The absolute values of $x$-Accuracy across $x$ are higher on the train set, as expected. The overall Black Scholes accuracy on Kinetics Set 2, Kinetics Set 3, Diving-48 and UAV Human are 80.15, 88.08%, 29.38% and 62.04%, very similar to the corresponding Black Scholes accuracies on the test sets, again indicating the generality of the Finterp across the train and test sets.

We present the confidence analysis in Figure 10. The trends, across $x$ as well as between different confidence metrics, are the same as that of train set, again proving that the Finterp doesn't compromise on the (accurate class) prediction probability in an attempt to minimize the length of video observed.

### A.4 Ablation Experiments

We conduct ablation experiments on the variables of the Black Scholes model on Set 2 of Kinetics, which has classes with a good amount scene information as well as motion/ temporal information. The rate $r$, volatility $\sigma$ and time to expiry $T - i$ are defined parameters, with no scope for change. The spot price $C_{spot}$ and strike price $C_{strike}$ are modeled as a combination of gradient norm $G_{spot}$, $G_{strike}$ and proportion of video observed $x$. In the first ablation, we remove the gradient norms $G_{spot}$, $G_{strike}$ and use just the proportion of video observed $x$. Naturally, this results a heavy bias towards selecting $x = 0.1, 0.2$ as the Black Scholes

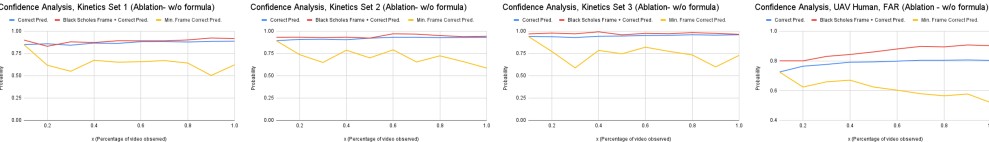

Figure 11: **Ablation experiments.** Analysis of the Black Scholes model for the ablation experiment where only prior cost $C_{spot}$ is used to find optimal frame, the Black Scholes formula is not used.

Figure 12: **Confidence analysis for ablation experiments.** Confidence analysis for the ablation experiment where only prior cost $C_{spot}$ is used to find optimal frame, the Black Scholes formula is not used.

frame which leads to trivial solutions. In the second ablation, we use just the gradient norm. This results in a low spot price $C_{spot}$ and high strike price $C_{strike}$ at all time instants, again creating a bias in the cost function. The result is that the Black Scholes Accuracy is $0.5\% - 7\%$ lower than the complete model at low time instants $x = 0.1 - 0.4$. The curve of Black Scholes Score % is also slightly shifted towards the right, indicating a preference towards low gradient time instants (or high values of $x$).

We next examine the impact of using the Black Scholes formula by finding the optimal time instant using only $C_{spot}$. In other words, we choose the time instant at which $C_{spot}$ is the lowest to be the optimal frame; the Black Scholes formula (involving $C_{strike}$, the cost at the final time instant) is not used. This corresponds to a trade-off between confidence at specific time instants and length of video observed, without taking into consideration the relative costs at the final time instant ($x = 1.0$). We conduct experiments on all three sets of Kinetics, Diving-48 and UAV Human. As shown in Figure 11, the graphs of Black Scholes accuracy are highly biased towards the right, optimal frame predictions are at much higher time instants. This is because the reliance on the confidence of prediction is large, high confidence predictions are highly prioritised. The model tends to prefer to use large proportions of the video to make action prediction despite not obtaining very large gains in confidence of prediction as compared to the Black Scholes time instant. The Black Scholes model uses the cost at the final time instant ($C_{strike}$) as well to find the optimal time instant, which ensures that the confidence of prediction is at the optimal time instant is not very far from the confidence from prediction at the final time instant. Note that there are two terms in the cost function for $C_{strike}$ anf $C_{spot}$, the first penalizes only the gradient, the second penalizes the gradient and the length of video observed simultaneously. Hence, the confidence of action prediction at optimal time instant is not comprised upon. Naturally, when optimal frames are concentrated towards high values of $x$, the Black Scholes accuracy will be very high (close to $90\% - 100\%$!). This is just a direct consequence of the $x$-Accuracy being high at high values of $x$, which makes these optimal frame predictions (with just $C_{spot}$) trivial solutions - the network needs to observe a large potion of the video. Rather, as stated earlier, it is beneficial to use the insights about optimal time instants gained from the Black Scholes model to enable neural networks to learn distinct temporal signatures for various actions in tasks such as action recognition, fine-grained action recognition, early action recognition, online action recognition, future prediction, etc. We present the confidence analysis for this ablation experiment in Figure 12. As expected, the confidence of accurate optimal frames for the ablation without Black Scholes formula is higher than the confidence of Black Scholes optimal frames computed using the Black Scholes formula. But again, the difference in the confidence scores in the two cases is not as significant, using a much smaller portion of the video to make an action prediction with slightly lower confidence is preferable.

### A.5    A note on comparisons with Transformer architectures

Analysis on the transformer-based video recognition architecture, TimeSformer Bertasius et al. (2021) result in the same findings as the other 3D CNNs we've used such as I3D and X3D. Our hypothesis for this is that 3D CNN backbones such as I3D and X3D as well as video transformer backbones such as TimeSformer behave similarly, i.e., they encapsulate knowledge across the spatial as well as temporal dimension in every step of computation in a unified manner.

### A.6    Comparisons with prior art

To the best of our knowledge, there is no prior work that attempts to study the earliest instant of time at which a pretrained action recognition neural network is capable of predicting the action class accurately. Early action recognition methods develop architectures that predict the action class using a partially observed video. Our goal is not to provide a solution for early action recognition, rather, it is to analyze the prediction capabilities of a pretrained action recognition neural network.

