# OpenReview forum: "Finterp: Cost-Time Analysis of Video Action Recognition using the Black Scholes Model"
_TMLR — Rejected by TMLR_

### Review · Reviewer_ddgM · 2023-07-14

**Summary Of Contributions:**

The paper proposes a cost-time analysis for video action recognition. The goal is to find the earliest instant at which the action depicted in a video can be predicted with high confidence. The main contribution of the paper is to adapt the Black Scholes model in finance to perform this analysis, resulting in a method called Finterp.

The Black Scholes model is a parametric model defined by an equation (Eq. 1 in the paper). Finterp redefines the meaning and value of each variable in this model to adapt it to video recognition. The authors use multiple values derived from common 3D CNN models for video recognition to define some of these variables.

Once this is done, the authors report different metrics derived from the model and interpret the obtained results. They show that different datasets have different cost-time characteristics and that predicting hierarchical actions (groups of similar actions) can be usually done at earlier times.

**Audience:**

Yes

**Broader Impact Concerns:**

No concerns.

**Claims And Evidence:**

No

**Requested Changes:**

- Conduct an experiment with a ViT-based action recognition model and comment on the differences between this model and 3D CNNs.


**Strengths And Weaknesses:**

[+] Novelty

The idea of performing a cost-time analysis of video recognition methods is novel and can provide interesting insights into these models.

[+] Thorough experimental section

The experimental section provides multiple experiments with several models on different datasets. The experiments highlight differences between the different datasets, including predictability of the actions and optimal time to do so.

[-] Black Scholes model assumptions might not apply to video

The Black Scholes model assumes certain conditions of the stock market, listed in Section 3.2. It is unclear that these assumptions hold for video recognition. In particular, it assumes that market movements are random, which is not quite true of videos - the future is a highly predictable function of the past (otherwise predicting future video frames from a few context frames would be impossible, and it is an active area of research).

[-] It is hard to assess whether Finterp makes sense

To define Finterp the authors define some variables of the Black Scholes model using properties (activations, gradients) of 3D CNN models. Finterp then shows that, in some datasets, the video action can be recognized quite early, whereas in other datasets the action is often not predictable with high accuracy (<50%). But it is hard to understand the validity of this analysis because of entangled factors: i) how accurate is the Black Scholes model to action recognition, and ii) the capabilities of the 3D CNNs used. Some kind of ablation study where other types of models (e.g. ViT-based models) are used would provide more insights to this aspect.

[-] The model is costly

Conducting the cost-time analysis using Finterp is quite computationally expensive, as it involves evaluating every possible video slice starting from the first frame (so frames[0:t], for all possible t) under the action recognition model and for all videos.

- General conclusion

The cost-time analysis model presented is novel, and the authors extract useful insights from it. However, it is unclear to me whether it makes sense to use such a model, designed for the stock market and with certain assumptions, to analyze video. Performing the cost-time analysis is costly, which will make it difficult to become widespread. Overall I see potential for this kind of analysis, but I am also not fully convinced of the validity of this approach.

---

### Review · Reviewer_T7Dt · 2023-07-16

**Summary Of Contributions:**

This article proposes a novel way of detecting the earliest instant of an action over a video sequence. The intuitions are given to analogize asset price with early action. The decreases in CNN gradient over consecutive frames are taken as price changes towards true price. Some analogies and assumptions are made to turn a pre-trained CNN model's outputs directly applicable in action detection.

**Audience:**

Yes

**Claims And Evidence:**

No

**Requested Changes:**

Clarify above points.
Please provide a clear function of Eq(1,2) with analogized video early detection variables in 3.3.2.

**Strengths And Weaknesses:**

Strengths:
- The intuitions are given to analogize asset price with early action appeared in a video sequence.
- Metrics of BS model are analyzed over three typical video recognition datasets.


Weakness:
- The terms of  "cost-time analysis"， "earliest instant of time" are coined in a strange way. They seem to relate to some new concepts, yet belongs to some extensively studied topic such as early action recognition [1]. More papers need to be cited and compared. What's the difference between this work and previous work?

[1] Michael S. Ryoo. Human Activity Prediction: Early Recognition of Ongoing Activities from Streaming Videos. 2011.


- The mimic of financial sequence study to early action recognition is not well-founded. Why does the video frames satisfy the Brownian motion conditional assumptions? Intuitively, video frames are sequentially progressed which impose strong frame-to-frame dependency. The assumptions introduced in 3.2 are related to financial asset, not sure how does this relate to video analysis. What are the implications of Assumption 1-4 to videos?

- Assumptions No.5  Figure not assigned.

- The assumption made in 3.3.1 requires the gradient decreases as the frozen CNN sees more frames as the confidence increases. However, how would this be ensured using a general purpose classification CNN?

- The linear decomposition of gradient score Gx * (1+x) is another assumption made about the behavior of the underlying CNN model.

- The rigorous formulations of a BS model over the defined G, sigma, v variables are obviously missing in 3.3.2. What cab the risk free rate be taken as the softmax prob of the CNN output?

- Comparisons of this approach to baseline methods are missing in experiment part. How would state-of-the-art AI methods perform on these three datasets?

---

### Review · Reviewer_agm1 · 2023-09-01

**Summary Of Contributions:**

This paper proposes a model for temporal analysis of videos, specifically finding the 'optimal' cost-time point, that is the earliest time at which the model can predict with 'high confidence' what the action is.

For this, the paper proposes to use the Black Scholes model from econometrics, and specifically for modelling stock price, call options, etc. The paper makes an analogy between the variables that are in the Black Scholes model (stock price, strike price, etc) and video-specific variables.

Then, after running the model on datasets, it assesses the predictions with respect to the cost-time dimension using the Black-Scholes model. That is, and as far as I understand, the paper makes an exploratory analysis of trained models on existing datasets and tries to understand how early models can predict reliably actions.

The experiments take place on Kinetics, Diving-48, and UAV Human.

**Audience:**

No

**Broader Impact Concerns:**

No.

**Claims And Evidence:**

No

**Requested Changes:**

I am afraid that the requested changes are simply to rewrite the paper taking into account the points in the weaknesses above.

**Strengths And Weaknesses:**

*Strengths*

- Understanding temporality in videos in the wild is crux for any future, envisioned generalist AI.
- The paper tries to make a contribution via a relatively unexplored angle, using econometrics/statistical models to derive conclusions.

*Weaknesses*

- **Unclear research hypothesis and technical description**. While the basic idea obviously hold merits, I have serious concerns with the extent with which one can find parallels between econometrics variables and video variables.
  - What is gradient score? I do see it being textually define afterward, but why not just write the equation. More importantly, the term `gradient score` does not really exist and it does sound like the score function, which is a different quantity. Why not call it gradient norm or gradient block norm?
  - Why is the gradient score a decreasing function? If the second derivative of the confidence of prediction is positive, wouldn't that mean that the gradient score is an increasing function? How do you know that the second derivative of the confidence is not positive?
  - Is the method only applicable in a 'transductive' setting, when one already has the ground-truth available? I am asking because in 3.3.2 we read the sentence `corresponding to the ground-truth mask`, which is normally not available at test time. If yes, the conclusions are mainly explanatory for the dataset, rather than specific for individual videos.
  - In 3.2 the assumptions do not really look convincing. Why would the 'true prediction' of a video require the whole video? The video is a data storage format. One can easily make it trivially longer and in that case a true prediction does not need the entirety of the video. Also, there are certain actions for which one can make a prediction without seeing it all, eg due to the scene bias as the paper posits. Further, I would say the other assumptions are quite shaky as well, but for the time being I will keep to this.
  - While I am not completely certain, I believe the implicit assumption in this work is that the video is short and contains the entirety of a single action and that one only. That is an extremely strong assumption, which also is incompatible with the task of detecting the earliest time point when an action happens. For this, wouldn't we want videos in the wild with random segments before and after?

- **Lack of clarity in writing**. The paper in its current form is full of unclear and vague statements. This not only makes the reading hard, but also renders unclear whether the model is actually doing anything reasonable or the method just returns some numbers, as anyways one would expect from a pretrained model. Exampels of unclarity follow:
  - Terminologies are mixed between finance and CS. For instance, the term cost function has a particular meaning in both fields. The way it is used now is quite confusing.
  - Quite significant use of finance terms, which are likely not familiar to a CS audience (time expiry, cost function, ...)
  - Using the variable $$x$$ for time rather than the commonplace $$t$$ is strange.
  - What is an `optimal frame`?
  - There exist missing figures (page 3 Figure ??)

- **Unconvincing results**.
  - To be honest, I am not really sure what to make out of the results. It does makes sense that with more frames (over time) prediction confidence increases.
  - I would also argue that the insights are unsurprising. Scene bias does make it very easy for the model to recognize the action, as the scene typically does not change and one can recognize it even from the first couple of frames. Also, motion bias is, I would say, a rather confusing term/concept, which I actually disagree with. Either way, it does makes sense that in the absence of any scene bias, and given enough granularity of action categories, at least a significant percentage of frames -say half of them- would be needed.
  - I think results would either way be more convincing with long and complex action datasets, such as EPIC Kitchen or perhaps even better, the Something-Something or Moment-in-Time.

---

### Comment · Reviewer_agm1 · 2023-09-09
**Thanks for the rebuttal**

I have gone through the rebuttal and the updated manuscript. I believe the changes are on the surface, and the main core of criticism (1. unclear motivation on the relevance of the optimal earliest point only on seen data, 2. 'forced' association between an econometrics model and a video understanding ML model) is still very much there. The authors answer that this is not their focus and it is interesting future work. I believe that then the work must be resubmitted in the future. Right now, the conclusions are mostly related to data analytics, and at least the predictive performance of the model must be validated to new unseen videos before acceptance to a journal like TMLR.

My recommendation remains unchanged.

---

### Comment · Reviewer_T7Dt · 2023-09-09
**Still need revisions in motivation descriptions and justification**

Thanks for the rebuttal.

I think I am still not convinced with the rebuttal that "the action that is executed in the (i+1)th frame is random, conditioned on the ith frame".  In a long run, given the complete history of 1-i frames, the frame of i+1 is pretty much certain. The stock market might be following the Brownian motion, since external actions are forcing upon the market. However, for structural data such as videos and sentences, based on  Markov property, the history can be condensed in previous frame and the next frame (event) follows some prob. distribution.

I am not sure if the efforts in connecting a financial concept with video understanding is justifiable, or necessary. I suggest just explicitly expressing your assumptions and suitable scenes, and more descriptions of the model itself with task.

---

### Comment · Reviewer_ddgM · 2023-09-11
**Thanks for the rebuttal, still unconvinced of the validity of the approach.**

Thank you authors for the rebuttal. After having gone through the rest of the reviews and the rebuttal, I am still unconvinced of the validity of the approach proposed in the paper.

Similarly to the other reviewers, I disagree with the assumption that the value of future frames in a video is completely random for all possible frames. Furthermore, I also agree with reviewer agm1 that the research hypothesis of the paper is unclear, and that the association with the BS model is forced because of the assumptions made.

Given this, my opinion of the paper remains unchanged.

---

### Author Response · Authors · 2023-09-17
**Clarification on assumptions**

We thank the reviewers for their time and valuable feedback to our rebuttal.

All reviewers raised concerns on the forced connection between financial markets and video frames. Our motivation for forging a connection between financial markets and video frames arises from the hypothesis that video frames satisfy certain assumptions, particularly, the Markovian assumption. The Markovian property of video frames has been used in video prediction and autoregressive generation in the past. This motivates us to exploit the Black Scholes formula to perform cost-time analysis of video frames, the hope is that this formulation (as well as the insights derived by the paper) will be useful in the development of video recognition models with unique temporal signatures for various actions.

[1] Yushchenko, V., Araslanov, N., & Roth, S. (2019). Markov decision process for video generation. In Proceedings of the IEEE/CVF International Conference on Computer Vision Workshops (pp. 0-0).

[2] Hoogeboom, E., Gritsenko, A. A., Bastings, J., Poole, B., Berg, R. V. D., & Salimans, T. (2021). Autoregressive diffusion models. arXiv preprint arXiv:2110.02037.

---

### Decision · Action_Editors · 2023-09-19

**Recommendation:** Reject

**Comment:**

All three expert reviewers recommended rejecting the current version of the paper. The AE has read through the paper, the reviews, response to the reviews, the revision, the discussion, and final recommendations. While the final recommendation comments from the reviewers are not public, they are far shorter summaries of the public comments posted by the reviewers.

After considering all of the inputs, the AE is inclined to agree with the reviewers that the manuscript needs a major revision to address multiple concerns.  The AE's options are accept as is, accept with minor revisions, and reject. Given these options, the AE is inclined to reject the paper but comment that the authors may want to consider a resubmission of a major revision. This major revision would require a more in-depth overhaul of the manuscript and its experiments than is feasible in a short revision.

The primary concerns of the reviewer are summarized well by the reviewers' final comments. To explain the decision and also potentially help the authors with a revision, the primary concerns underlying the decision are twofold.

### Assumptions and connections to econometrics

The reviewers thought that connection between the econometrics model and the ML model were not made sufficiently clear. This was a general concern, with a few different specific concerns raised by each reviewer. One key sticking point is the connection to the Black-Scholes model, which is a central part of the paper. All three reviewers raised or later commented on concerns about the connection.

In the Black-Scholes model, the the stock price is assumed to follow geometric Brownian motion. The reviewers did not believe that videos follow geometric Brownian motion.  While the authors have argued that future frames are random, the reviewers remain unconvinced.

To summarize and interpret the discussion, both the authors and reviewers agree that, when conditioned on the ith frame, the i+1th frame follows  some distribution. The disagreement comes from what that distribution is. Brownian motion has extremely stringent requirements: continuity, independence from frame to frame, and normality. These are not obviously true for the probability of an action (or pixels or flow) over time.

In a revision, the authors may want to make the assumptions crystal clear and show either:
- that the data follows the Brownian motion model well in some space (optical flow, latent feature space, pixels) on some data;
- why deviating from the model doesn't impact results, at least on the analyzed data.

### Clarity of research hypothesis and contribution

The reviewers have also raised concerns about clarifying the specific research hypothesis pursued, the motivation, and the potential application.

Interpreting and summarizing the discussion and reviewer comments, these concerns can likely be resolved by some subset of:
- *Showing the effectiveness of Finterp as an analysis method compared to some simple baselines.* The authors could show that Finterp produces more accurate result than some other straightforward approaches. It is true that the work differs from other past work in the sense that it analyzes pre-trained models. However, in this setting, there may be easy heuristics that can be compared with.
- *Showing that the analysis helps identify important data issues.* The reviewers described some of the outcomes as obvious (more frames = better), while the authors responded that the point of the method is to find the amount of time that is needed. The authors may be able to address this by using Finterp to obtain important insights into the data (e.g., at the class level, or on more data as requested by reviewers), and then show that these insights translate into a practical, empirical improvement.

**Audience:**

Yes, although the presentation of the results may need some modification. Please see the second comment in the discussion under the recommendation. There were concerns about the clarity of the specific research hypothesis and baselines.

**Claims And Evidence:**

Partly, but not clearly. Please see the discussion under the recommendation. The reviewers thought that the current manuscript does not make the connection between Black-Scholes and the proposed analysis clear.

**Resubmission Of Major Revision:**

The authors may consider submitting a major revision at a later time.